# OpenReview forum: "Data Valuation by Leveraging Global and Local Statistical Information"
_NeurIPS.cc/2025/Conference — Submitted to NeurIPS 2025_

### Official Review · Reviewer_92n8 · 2025-06-16

**Clarity:** 3
**Significance:** 3
**Originality:** 3
**Rating:** 4
**Confidence:** 4

**Summary:**

The paper builds upon an existing work that proposes how to efficiently approximate the Average Marginal Effect (AME) such as the Shapley value. Instead of the original approach that uses the solution to Lasso Regression, the paper proposes to modify the objective to include (1) a different global term and (2) a new local term. (1) uses the L2 norm based on the observation that data values tend to follow a Gaussian distribution instead of a Laplace distribution. (2) is designed to enforce that data with similar features and labels should have more similar data values. Next, the paper proposes adapting the objective to deal with dynamic data valuation where new data points are added or removed. Instead of recomputing the AMEs, the objective is to keep the data values within some permissible variation decided by heuristics. The paper empirically compares the proposed method, GLOC, against the original AME approach and other data valuation techniques on multiple datasets and settings (mislabeled data detection, dynamic data valuation).

**Questions:**

1. Can you better explain Fig 1c-d? What dataset is used? Do you consider the pairwise difference in data values? Is there any significant difference between the blue and orange line?
2. Can Fig 3 be modified to compare the MSE and MAE achieved by the original regression problem with no regularisation, AME, GLOC, GLOC without R_g and GLOC without R_l? Is the graph similar across different sample size? Is it meaningful to compare the Spearman correlation between each method's data value and the exact SV?
3. Does GLOC still outperform AME when removing data points with _low_ values and adding those with _large_ values?
4. (Limitations of GLOC) On what datasets and data samples do GLOC underperform? Is GLOC also bad at assigning low values to mislabeled data?

**Ethical Concerns:**

["NO or VERY MINOR ethics concerns only"]

**Final Justification:**

The authors' rebuttal and new experiments address all of my concerns but the original score was high enough.

**Limitations:**

The limitations are discussed in App A.12.

**Quality:**

3

**Strengths And Weaknesses:**

### Strengths
1. The experiments show that GLOC significantly improves the solution over AME and the heuristics proposed (e.g. L2 regularisation) are justified empirically.
2. It is novel to consider adapting GLOC for dynamic data valuation to avoid the need for reestimating Shapley values.

### Weaknesses
1. Some claims can be made clearer.
    * For example, in line 111 of the related work, it is stated that "existing algorithms largely disregard the distributional characteristics of data values, leading to suboptimal performance and efficiency. On line 114, "distributional information" is mentioned. However, to the reader, the idea and importance of distributional information is unclear as it is not explained. Based on the experiment section, it seems that GLOC is also marginal contribution based and is able to obtain a solution with lower MSE based on different assumptions about the data value distribution.
    * Clearer explanation of Fig 1.c-d
2. The design choices are based on heuristics and empirically validated instead of theoretical motivations (e.g. LASSO for sample efficiency). Thus, there is no theoretical guarantees.

### Minor Comments
* line 91: "Sharpley" -> "Shapley"
* In Fig 1, clearly state the null hypothesis.
* cite the specific appendix

---

> ### Author Rebuttal · Authors · 2025-07-31
>
> Dear Reviewer 92n8,
>
> We sincerely appreciate your thorough review and insightful comments and suggestions. We greatly thank you for your recognition of the soundness and novelty of our method, the comprehensiveness of our experimental design, and the strong performance demonstrated in our results. Below are our detailed responses to your concerns.
>
> > **Q1: Explanation for “distributional information”.**
>
> To further clarify the idea and significance of distributional information, we have revised the sentences in our manuscript as follows:
> - Line 111: “*However, existing algorithms largely disregard the distributional information of data values, leading to suboptimal performance and efficiency. Distributional information, encompassing both global and local distribution characteristics, enables the capture of structural relationships among samples, evaluation of sample importance and representativeness, and guidance of model training and regularization. Such aspects have been applied across various machine learning tasks and will be discussed in detail in the following subsection.*”
> - Line 114: “*In machine learning, various methods have incorporated both global and local distributional information during model training [35, 45]. Specifically, the global distribution refers to assumptions about the overall data distribution (e.g., Gaussian or Laplace), whereas the local distribution pertains to the distribution of samples within a neighborhood or localized region.*”
>
> Existing marginal-contribution-based methods evaluate each data point independently, relying solely on random subset sampling. In contrast, GLOC leverages both global and local distributional information to refine Shapley value estimation, yielding more accurate valuations and reduced MSEs. Although we primarily demonstrate the effectiveness of our approach in combination with AME within the standard data valuation scenario, **our method is general and can be easily integrated with other valuation techniques**. Specifically, the proposed distribution-aware regularization terms can be applied either jointly with the original method or as a post-processing step (**Section 4.4**). Additional results are provided in **Table 4**.
>
> > **Q2: Clearer explanation of Figs. 1(c) and (d).**
>
> Figs. 1(c) and (d) display results based on the **Random dataset (Eq. (12))**, illustrating the average relative difference between a sample's value and those of its neighbors, both within the same (c) and different (d) classes. In both figures, the blue curve represents a sample from the positive class (+1), and the orange curve represents a sample from the negative class (–1). A notable pattern emerges: **samples within the same class exhibit smaller value differences with neighboring samples, while inter-class proximity often results in larger differences**. Results for two additional randomly selected samples, shown in the table below, align with this trend.
>
> Same category:
> |Range|0.2|0.3|0.4|0.5|0.6|0.7|
> |:-|:-:|:-:|:-:|:-:|:-:|:-:|
> |Class +1|0.26|0.37|0.46|0.52|0.54|0.55|
> |Class -1|0.35|0.43|0.48|0.52|0.53|0.54|
>
> Different category:
> |Range|0.2|0.3|0.4|0.5|0.6|0.7|
> |:-|:-:|:-:|:-:|:-:|:-:|:-:|
> |Class +1|0.75|0.70|0.68|0.65|0.62|0.61|
> |Class -1|0.78|0.72|0.66|0.64|0.63|0.60|
>
> In response to your valuable feedback, we have revised the figure caption to provide more specific details: “*The average relative difference after min-max normalization between the value of a sample and the values of its neighbors within the same (c) and different (d) categories. The Random dataset defined in Eq. (12) is utilized. The blue curve represents a sample from the positive (+1) class, and the orange curve denotes a sample from the negative (–1) class.*”
>
> > **Q3: Theoretical guarantees.**
>
> Our main objectives are to address two key challenges in data valuation: 1) improving Shapley value estimation accuracy and 2) enhancing the accuracy and efficiency of dynamic data valuation. While not solely theory-driven, each design component is supported by strong motivations and empirical evidence.
> - Previous studies rely on the sparsity assumption of data values and use L1 regularization, without explicitly considering the underlying value distribution [1]. While L1 regularization implicitly assumes a Laplace prior, our experimental analysis shows that real-world data values are more closely aligned with a Gaussian distribution. Hence, **we adopt L2 regularization, which is theoretically linked to a Gaussian prior, to improve valuation accuracy**.
> - The local regularization term is based on the empirical observation that adjacent data points within the same class exhibit substantial value similarity, while those from different classes show minimal similarity. This aligns with the **manifold assumption**, which posits that similar samples in local neighborhoods share similar representations and properties.
>
> Although our work lacks formal theoretical guarantees, such as convergence proofs or error bounds, we provide extensive evaluations across various datasets and tasks, including Shapley value estimation, data valuation for addition and removal, mislabeled data detection, and dynamic data valuation. **The results show that our method consistently outperforms existing baselines.**
>
> Following your valuable comments, we have revised the limitations section to include the following sentences: “*The proposed method is primarily empirically driven. While it demonstrates strong performance across various tasks, it currently lacks theoretical guarantees, such as convergence and error bounds. Addressing these aspects will be the focus of our future work.*”
>
> > **Q4: Minor comments.**
> - We greatly appreciate your thorough review and corrections. We have corrected the spelling errors and carefully reviewed the manuscript to avoid similar issues.
> - The null hypothesis-that data values follow a Gaussian distribution-has been clarified in the captions of Figs. 1(a) and (b): "*The KStest is performed under the null hypothesis that the data values follow a Gaussian distribution.*"
> - All appendix references have been updated to cite specific sections, figures, and tables for improved clarity.
>
> > **Q5: Extensive comparison of Fig. 3.**
>
> We humbly respond to your concerns from the following two aspects:
> - Inspired by your valuable feedback, we have expanded the comparisons in Fig. 3 to include both **MSE and MAE**, as shown in the table below. **When one regularization term is removed, GLOC's performance declines but still consistently outperforms AME, highlighting the necessity of each regularization term**. Moreover, despite the significant difference in sample sizes between Electricity (38,474 samples) and BBC (2,225 samples), the performance trends remain consistent.
> |Dataset|Metric|AME|GLOC|$-\mathcal{R}_{g}$|$-\mathcal{R}_{l}$|
> |:-|:-:|:-:|:-:|:-:|:-:|
> |Electricity|MSE|4.16e-5|**8.33e-7**|9.56e-6|1.41e-5|
> ||MAE|5.23e-3|**7.30e-4**|2.47e-3|3.00e-3|
> |BBC|MSE|1.06e-5|**1.76e-6**|5.82e-6|6.51e-6|
> ||MAE|2.61e-3|**1.06e-3**|1.92e-3|2.01e-3|
> - Excellent suggestion! **Spearman correlation complements MSE/MAE-based evaluation by capturing consistency in relative value rankings**. The table below shows the Spearman correlation coefficients between data values estimated by different methods and the approximated ground truth. Notably, **GLOC estimates show the highest correlation with the ground-truth Shapley values, further validating its effectiveness in achieving accurate data valuation**.
> |Dataset|AME|GLOC|$-\mathcal{R}_{g}$|$-\mathcal{R}_{l}$|
> |:-|:-:|:-:|:-:|:-:|
> |Electricity|0.68|**0.85**|0.80|0.75|
> |BBC|0.72|**0.81**|0.75|0.77|
>
> > **Q6: Comparison between GLOC and AME when removing data points with low values and adding those with large values.**
>
> **Given the higher valuation accuracy, GLOC is more effective at distinguishing high-quality from low-quality samples**. The tables below compare GLOC and AME in two scenarios: removing low-value samples and adding high-value samples. Ideally, removing the least valuable samples or adding the most informative ones should improve model performance under noisy conditions. As shown below, **GLOC consistently outperforms AME across both scenarios**.
>
> Removing low-value samples for MiniBooNE and 2Dplanes:
> |Ratio|0.05|0.1|0.15|0.2|
> |:-|:-:|:-:|:-:|:-:|
> |AME|0.54|0.62|0.69|0.74|
> |GLOC|0.62|0.71|0.79|0.83|
>
> |Ratio|0.05|0.1|0.15|0.2|
> |:-|:-:|:-:|:-:|:-:|
> |AME|0.57|0.63|0.70|0.75|
> |GLOC|0.64|0.71|0.80|0.84|
>
> Adding high-value samples for Electricity and Fried:
> |Dataset|0.2|0.4|0.6|0.8|
> |:-|:-:|:-:|:-:|:-:|
> |AME|0.59|0.64|0.67|0.69|
> |GLOC|**0.61**|**0.67**|**0.71**|**0.73**|
>
> |Dataset|0.2|0.4|0.6|0.8|
> |:-|:-:|:-:|:-:|:-:|
> |AME|0.70|0.72|0.76|0.79|
> |GLOC|**0.77**|**0.80**|**0.84**|**0.86**|
>
> > **Q7: Limitations of GLOC.**
>
> Following your valuable comments, we demonstrate two extreme scenarios in which GLOC exhibits degraded performance.
> - A mislabeled sample may appear consistent within the feature space of its incorrect class, leading the local regularization term to erroneously assign it a higher value.
> - If a group of noisy samples forms a coherent but incorrect cluster—such as when an entire cluster is mislabeled—the local regularization may enforce similarity among their valuations, potentially resulting in a collectively overestimated value.
>
> Moreover, **GLOC overcomes AME's limitation in detecting mislabeled samples**. AME's inferior noise detection performance stems from its use of L1 regularization, which induces sparsity and assigns zero values to many samples, including non-noisy ones. In contrast, our method utilizes L2 regularization, based on the observation that data values follow a global distribution closer to Gaussian than Laplace. Additionally, local distribution-based regularization further suppresses noisy samples, improving noise detection.
>
> [1] Measuring the effect of training data on deep learning predictions via randomized experiments. ICML 2022.

---

> > ### Comment · Reviewer_92n8 · 2025-08-04
> >
> > Thank you for your detailed and helpful response! The response has addressed my questions and concerns about GLOC empirical performance.

---

> > > ### Author Response · Authors · 2025-08-04
> > > **Thank you for your acknowledgment.**
> > >
> > > Dear Reviewer 92n8,
> > >
> > > Thank you very much for your timely reply! We are delighted that you found our reply to be detailed and helpful, and that it successfully addressed your concerns. We are also sincerely grateful for your recommendation of the acceptance of our manuscript, which has greatly encouraged our enthusiasm for this line of research.
> > >
> > > We are fully committed to integrating all revisions discussed during the rebuttal period into the final version of our manuscript, as these changes have substantially improved the overall quality and clarity of our work. Once again, thank you for your thoughtful feedback and for your invaluable role in improving our work!
> > >
> > > Kind regards,
> > >
> > > Authors

---

### Official Review · Reviewer_Vwbk · 2025-06-25

**Clarity:** 3
**Significance:** 2
**Originality:** 3
**Rating:** 4
**Confidence:** 4

**Summary:**

The paper studies a new data valuation method considering the distribution of value and position information. Inspired by the difference between LASSO and Ridge, the authors derive a valuation with a quadratic penalty after observing a Gaussian-like distribution. The valuation based on value distribution is quite novel, and including similarity among data is natural and reasonable. However, the experimental design is not very convincing, and the literature review doesn't cover sufficient recent papers.

**Questions:**

Why do you use $||\beta||_2$ rather than $||\beta||_2^2$, as in Ridge regression, the penalty term is L2 norm square？How to choose $k$ in your experiments?

**Ethical Concerns:**

["NO or VERY MINOR ethics concerns only"]

**Final Justification:**

The reviewer's rebuttal has addressed most of my conerns, and provided thorough experiments for a few questions I had. I raise my rating by 1 point, though think the paper is not to a level of "accept" yet due to its somewhat incremental techniques.

**Limitations:**

Yes

**Quality:**

2

**Strengths And Weaknesses:**

Studying the value distribution and including the similarity is quite insightful. However, as one of the strengths of the new valuation is its computational efficiency, the paper lacks a comparison with existing methods.

On Lines 152-153, as you assume X comes from a Gaussian distribution, it's not very convincing to say the value distribution is Gaussian-like. For example, you could conduct another simulation with Laplace-like X and conduct the same test.

On Line 249, "limited range" seems not rigorous. The paper only shows 10% won't change the valuation significantly. Changing the proportion and monitoring the trend will be helpful to understand the actual influence.

Besides, the paper doesn't contain up-to-date literature. Most of the references are before 2024. For example, "The Value of Information in Human-AI Decision-making" and "An Instrumental Value for Data Production and its Application to Data Pricing" consider the dynamic case.

---

> ### Author Rebuttal · Authors · 2025-07-31
>
> Dear Reviewer Vwbk,
>
> We sincerely appreciate your thorough review and constructive comments. We are grateful for your recognition of the novelty and soundness of our method, as well as your positive appraisal of our insightful contributions. Below are our detailed responses to your concerns.
>
> > **Q1: As one of the strengths of the new valuation is its computational efficiency, the paper lacks a comparison with existing methods.**
>
> Thanks for your valuable comments. **Fig. 9** in our manuscript compares the computation times of IncGLOC and DecGLOC with existing baselines. The results show that **methods avoiding Shapley value re-estimation—such as KNN, KNN+, and ours—consistently achieve superior efficiency**. Moreover, the table below compares the computation time of the proposed GLOC method (based on AME) with the original AME. Results indicate that GLOC, which introduces only two distribution-aware regularization terms to AME, **incurs minimal additional training time while improving Shapley value estimation accuracy by up to 206 times** (Table 1).
> |Dataset|BBC|CIFAR10|Law|Electricity|Fried|
> |:-|:-:|:-:|:-:|:-:|:-:|
> |AME|21.45s|489.54s|7.24s|8.32s|10.48s|
> |GLOC|21.58s|490.13s|7.25s|8.34s|10.53s|
>
> > **Q2: As you assume X comes from a Gaussian distribution, it's not very convincing to say the value distribution is Gaussian-like. You could conduct another simulation with Laplace-like X and conduct the same test.**
>
> Constructive suggestion. Based on the results in Figs. 1 and 6, the data value distributions for CIFAR10, Random, BBC, 2Dplanes, Fried, and MiniBooNE exhibit Gaussian-like patterns. Notably, while the Random dataset is synthetically constructed, the others are real datasets and are not subject to any assumptions about Gaussian distributional form.
>
> In response to your valuable suggestion, we have replaced the Gaussian distribution in Eq. (12) with an independent Laplace distribution for each feature dimension: $y \stackrel{u.a.r}{\sim}$ \{-1,+1\}; $\boldsymbol{\theta}=[{+1, +1} ] ^ T \in \mathbb{R} ^ {2}$; $\boldsymbol{x} \sim \begin{array}{ll}{Laplace}(\boldsymbol{\theta}, b _ {+}),  \text { if } y=+1; \boldsymbol{x} \sim {Laplace}(-\boldsymbol{\theta}, b_{-} ),  \text { if } y=-1\end{array}$. We then repeat the Shapley value estimation and distributional analysis on this dataset. The KStest yields a p-value of **0.5815723456146522**, indicating that the resulting data value distribution remains consistent with a Gaussian distribution. This supports the conclusion that **the observed Gaussian-like pattern is not solely due to the assumption of Gaussian-distributed input features**.
>
> > **Q3: On Line 249, "limited range" seems not rigorous. The paper only shows 10% won't change the valuation significantly. Changing the proportion and monitoring the trend will be helpful to understand the actual influence.**
>
> We humbly respond to your concerns from the following perspectives:
> - Most existing dynamic data valuation studies consider only minimal modifications to the dataset, often limited to the addition or removal of one or two samples [1]. Under such small-scale changes, it is expected that the data value distribution remains largely unaffected. Our experiments (shown in **Fig. 2**) demonstrate that when removing or adding 10% of the dataset, the change in the distribution of data values remains small.
> - Nonetheless, we fully agree that a more comprehensive analysis across a wider range of modification ratios would provide deeper insight into the influence of dataset changes. To this end, we systematically increased the ratio of modified data from 5% to 40% in increments of 5%. The results presented below yield the following findings:
> |Ratio|Original|5%|10%|15%|20%|25%|30%|35%|40%|
> |:-|:-:|:-:|:-:|:-:|:-:|:-:|:-:|:-:|:-:|
> |Random|$\mu$=0.56, $\sigma$=0.22|$\mu$=0.56,$\sigma$=0.21|$\mu$=0.55,$\sigma$=0.21|$\mu$=0.53,$\sigma$=0.20|$\mu$=0.50,$\sigma$=0.21|$\mu$=0.50,$\sigma$=0.19|$\mu$=0.47,$\sigma$=0.25|$\mu$=0.44,$\sigma$=0.26|$\mu$=0.43,$\sigma$=0.25|
> |Electricity|$\mu$=0.44, $\sigma$=0.15|$\mu$=0.46,$\sigma$=0.15|$\mu$=0.49,$\sigma$=0.16|$\mu$=0.47,$\sigma$=0.17|$\mu$=0.41,$\sigma$=0.18|$\mu$=0.39,$\sigma$=0.18|$\mu$=0.52,$\sigma$=0.24|$\mu$=0.54,$\sigma$=0.22|$\mu$=0.57,$\sigma$=0.23|
> - **When the dataset undergoes modifications of up to 15%, the deviation in the distribution of data valuations remains minimal, staying within ±0.05 on a normalized scale.**
> - Once the modification exceeds 30%, the distribution of data valuations begins to exhibit more pronounced shifts.
>
> **Given that dynamic data valuation methods are primarily intended for scenarios involving relatively minor dataset modifications, our approach—which updates data values based on both the modified dataset and previously estimated values—is both effective and appropriate**. However, when substantial changes occur, it is more suitable to recompute data values using the proposed GLOC method. Alternatively, a step-wise strategy can be adopted, where no more than 10% of the dataset is modified at each step, and IncGLOC or DecGLOC is applied iteratively for incremental or decremental valuation.
>
> > **Q4: Most of the references are before 2024. For example, "The Value of Information in Human-AI Decision-making" and "An Instrumental Value for Data Production and its Application to Data Pricing" consider the dynamic case.**
>
> Thank you for your helpful suggestion. We have carefully considered the two papers you recommended:
> - “*The Value of Information in Human-AI Decision-making*” [2] investigates information complementarity in human-AI collaboration by quantifying the marginal utility of signals within a decision-theoretic framework. Its key contribution lies in providing a principled perspective on information gain, with ILIV-SHAP enhancing interpretability by identifying features with unique decision value. However, the work primarily emphasizes explanation and strategy design, and does not offer methods readily applicable to efficient Shapley value estimation or data-centric model performance analysis.
> - “*An Instrumental Value for Data Production and its Application to Data Pricing*” [3] proposes a theoretical framework for quantifying the instrumental value of data, highlighting its marginal contribution and applying it to the design of pricing and sales mechanisms. A key strength of this work is its alignment with the Shapley value intuition, offering a valuation method that does not require access to realized data. However, the framework relies on a pre-defined regression model and prior distribution, limiting its applicability to general-purpose machine learning models such as deep neural networks. Overall, the approach is more focused on economic modeling and mechanism design than on practical data processing workflows.
>
> In summary, **while the aforementioned studies provide valuable theoretical foundations and are well-suited to data pricing scenarios, their methods are not readily applicable to sample-level data manipulation tasks. In contrast, our approach offers greater flexibility, enabling the estimation of individual data points’ complementary contributions to existing models or experts [2], and can serve as a practical valuation module within the framework proposed by [3]**. These extensions suggest promising directions for future research. Moreover, both studies underscore the increasing importance of data and information valuation in both academic and practical contexts.
>
> Accordingly, we have included these studies in the **related work** section of our manuscript. The added content is as follows: "*In addition to data valuation methods developed for traditional machine learning, there are also studies focusing on broader applications of data valuation, such as Human-AI Decision-making and Data Pricing [42,44]. These studies not only highlight the critical importance and wide-ranging applicability of data valuation techniques but also provide a foundational framework for future efforts to extend our approach to more diverse domains, such as decision support systems and data marketplaces.*"
>
> > **Q5: Why do you use $|| \boldsymbol{\beta} || _ {2}$  rather than $|| \boldsymbol{\beta} || _ {2} ^ {2}$, as in Ridge regression, the penalty term is L2 norm square？How to choose $k$  in your experiments?**
>
> - Good question. We apologize for the oversight in our original manuscript. **Both our proposed method and theoretical framework are grounded in standard L2 regularization $|| \boldsymbol{\beta} || _ {2} ^ {2}$, which corresponds to a zero-mean Gaussian prior**. We have carefully reviewed the manuscript and corrected all occurrences of $|| \boldsymbol{\beta}|| _ {2}$ to the proper notation $|| \boldsymbol{\beta}|| _ {2}^{2}$. We appreciate your attention to this detail and thank you for bringing it to our attention.
> - **Section A.11 (Figs. 10(c) and (d))** evaluates the impact of neighborhood size ($k$) on IncGLOC performance. The tables below further present ablation studies of $k$ for GLOC and DecGLOC, reporting integer MSE ratios between AME and our approach. Consistent with the results obtained for IncGLOC, **optimal performance is observed at k=5 or k=8** for all three methods across various datasets, which can serve as recommended default settings.
>
> For GLOC:
> |$k$|3|5|8|10|15|
> |:-|:-:|:-:|:-:|:-:|:-:|
> |Electricity|27:1|**50:1**|46:1|45:1|30:1|
> |CIFAR10|53:1|**96:1**|95:1|87:1|48:1|
> |2Dplanes|58:1|**105:1**|**105:1**|101:1|64:1|
> |BBC|4:1|6:1|**8:1**|5:1|4:1|
>
> For DecGLOC:
> |$k$|3|5|8|10|15|
> |:-|:-:|:-:|:-:|:-:|:-:|
> |Electricity|39:1|**67:1**|65:1|59:1|45:1|
> |MiniBooNE|15:1|**21:1**|19:1|16:1|13:1|
> |CIFAR10|47:1|95:1|**96:1**|85:1|71:1|
> |Fried|3:1|**9:1**|**9:1**|7:1|5:1|
>
> [1] Dynamic shapley value computation. ICDE 2023.
>
> [2] The value of information in human-AI decision-making. arXiv preprint arXiv:2502.06152 (2025).
>
> [3] An instrumental value for data production and its application to data pricing. ICML 2025.

---

> ### Comment · Reviewer_Vwbk · 2025-08-04
> **Thanks for the Rebuttal!**
>
> I appreciates the authors' thorough response to my questions. The authors have addressed most of my concerns and also slightly raised my evaluation about the paper. I will raise my rating accordingly.

---

> > ### Author Response · Authors · 2025-08-04
> > **Thank you for your encouraging feedback!**
> >
> > Dear Reviewer Vwbk,
> >
> > Thank you very much for your timely reply! We are truly grateful that you found our responses to be thorough and that they have addressed your concerns. We also sincerely appreciate your decision to raise your score—this means a great deal to us and serves as a strong encouragement.
> >
> > We are fully committed to incorporating all the revisions discussed during the rebuttal phase into our final manuscript, as these changes have greatly contributed to improving the quality and clarity of our work. Once again, thank you for your valuable feedback and for the important role you have played in helping us strengthen our paper!
> >
> >
> > Kind regards,
> >
> > Authors

---

### Official Review · Reviewer_58ya · 2025-07-01

**Clarity:** 3
**Significance:** 3
**Originality:** 3
**Rating:** 5
**Confidence:** 3

**Summary:**

The authors propose AME-based GLOC, IncGLOC, and DecGLOC data valuation methods based on the identified global and local distribution characteristics of data values.
They theoretically define the methods and algorithms and empirically evaluate them on several real-world datasets, comparing the results to those of 11 existing data valuation methods.

**Questions:**

Below are the questions, clarifications and concerns I would like the authors to address

- Section 3
   - Clarifications about Figures 1c and 1d.
       - Are the results on the random dataset?
       - Is there an explanation for why class +1 is always higher?
       - Is there a class imbalance?
       - If the difference is truly in % (that is, if it's not a mistake), then those values for both inter and intra are very low.
   - On line 152, to improve readability, I think that authors should include where in the appendix more results on this are presented.
   - As discussed in Section 3, the authors appear to use the AME-based SV estimator. Instead of only mentioning AME, I think it would improve readability to say AME-based SV estimator because it can be confusing to see Shapley values and then AME.


- Sections 4.2 and 4.3
   - How does the choice of Nk (line 220) and the similarity metric affect things?
   - Since the CS-Shapley data valuation method discriminates between training instances' in-class and out-of-class contributions, can the authors compare GLOC to Class-Shapley (Schoch et al. 2022)?
   - Since most marginal contribution methods are not stable across runs, are affected by choice and design of utility function, and are in most cases not rank stable, is it possible that reliance on current values \Beta^cur might lead to over or under-estimation of values for the added/removed data points?
   - Since one of the mentioned advantages of the method is computational efficiency, can the authors add a computational analysis (runtime analysis) of the three proposed methods?

- Section 5
  - The authors say, "Given the benchmark Shapley values (SV)," but it's unclear which of Shapley methods among those listed they use and how they compute the values. Also, why are Shapley values used as the benchmarks?


- Other observations, questions, and concerns
   - The likely limitations of the proposed framework.  1) To estimate Shapley values under L2 error, the monotonicity and sparsity assumptions must hold, which may not always be the case, depending on the data distribution and utility section, among other design choices. 2) Hyperparameter tuning Nk could make the proposed data valuation methods harder to scale. 3) Reliance on neighbors might compromise privacy guarantees. 4) There is a possibility of (over)underestimating the value of some samples. 5) There is a possibility of a mismatch between the underlying model, resulting utility, and the valuation method. 6) Since the efficiency axiom does not hold in this case (according to [25]), then it might be hard to guarantee the nice "fairness" properties.

**Ethical Concerns:**

["NO or VERY MINOR ethics concerns only"]

**Final Justification:**

The authors have sufficiently addressed all questions we had. I have therefore decided to raise my score, with the expectation that the authors incorporate the new evaluations and all the promised changes into the paper.

**Limitations:**

Authors have a limitations section in the appendix. However, I think there are several limitations, e.g., those in the added in the "questions" section above that authors should consider adding or discussing.

**Quality:**

3

**Strengths And Weaknesses:**

**Strengths**

- The Authors identify interesting distributional patterns of data values and leverage the AME method to propose a global and local characteristics-based (GLOC) data valuation approach and two dynamic data valuation methods (IncGLOC and DecGLOC) designed for scenarios involving the addition of new data and the removal of existing data.

- Figure 1 and 2 presents interesting insights, and the authors' perspective on dynamic data valuation is particularly relevant and could be highly valuable in the context of data markets.

- The authors conducted experiments on several datasets, compared their work with several data valuation methods, and also included their code in the supplemental material. (I haven't tested it yet)

- In general, the paper is well-written and easy to understand.


**Weaknesses**

- Section 3
   - Clarifications about Figures 1c and 1d.
       - Are the results on the random dataset?
       - Is there an explanation for why class +1 is always higher?
       - Is there a class imbalance?
       - If the difference is truly in % (that is, if it's not a mistake), then those values for both inter and intra are very low.
   - On line 152, to improve readability, I think that authors should include where in the appendix more results on this are presented.
   - As discussed in Section 3, the authors appear to use the AME-based SV estimator. Instead of only mentioning AME, I think it would improve readability to say AME-based SV estimator because it can be confusing to see Shapley values and then AME.


- Sections 4.2 and 4.3
   - How does the choice of Nk (line 220) and the similarity metric affect things?
   - Since the CS-Shapley data valuation method discriminates between training instances' in-class and out-of-class contributions, can the authors compare GLOC to Class-Shapley (Schoch et al. 2022)?
   - Since most marginal contribution methods are not stable across runs, are affected by choice and design of utility function, and are in most cases not rank stable, is it possible that reliance on current values \Beta^cur might lead to over or under-estimation of values for the added/removed data points?
   - Since one of the mentioned advantages of the method is computational efficiency, can the authors add a computational analysis (runtime analysis) of the three proposed methods?

- Section 5
  - The authors say, "Given the benchmark Shapley values (SV)," but it's unclear which of Shapley methods among those listed they use and how they compute the values. Also, why are Shapley values used as the benchmarks?


- Other observations, questions, and concerns
   - The likely limitations of the proposed framework.  1) To estimate Shapley values under L2 error, the monotonicity and sparsity assumptions must hold, which may not always be the case, depending on the data distribution and utility section, among other design choices. 2) Hyperparameter tuning Nk could make the proposed data valuation methods harder to scale. 3) Reliance on neighbors might compromise privacy guarantees. 4) There is a possibility of (over)underestimating the value of some samples. 5) There is a possibility of a mismatch between the underlying model, resulting utility, and the valuation method. 6) Since the efficiency axiom does not hold in this case (according to [25]), then it might be hard to guarantee the nice "fairness" properties.

---

> ### Author Rebuttal · Authors · 2025-07-31
>
> Dear Reviewer 58ya,
>
> We sincerely appreciate your insightful review and valuable comments. Thank you for your recognition of our work as both interesting and valuable, our paper as well-written and easy to follow, our experiments as comprehensive, and our code as readily available. Below, we provide our detailed responses to each of your concerns.
>
> > **Q1: Clarifications about Figs. 1(c) and (d).**
>
> We apologize for the unclear explanation of Figs. 1(c) and (d), and provide the following clarification:
> - As you correctly specified, the results in Figs. 1(c) and (d) are based on the **Random dataset (Eq. (12))**, which introduces class imbalance through varying category difficulty, modeled by assigning a higher variance to class +1.
> - The two figures show the average relative difference between a sample's value and those of its neighbors within the same (c) and different (d) classes. The blue or orange curve represents a positive (+1) or negative (-1) class sample. **The observed tendency for class +1 to exhibit higher differences does not represent an inherent pattern, which varies with sample selection**, as shown in the table below using two random alternative samples. Moreover, **all differences are min-max normalized** to enhance clarity and presentation.
>
> Same category:
> |Range|0.2|0.3|0.4|0.5|0.6|0.7|
> |:-|:-:|:-:|:-:|:-:|:-:|:-:|
> |Class +1|0.26|0.37|0.46|0.52|0.54|0.55|
> |Class -1|0.35|0.43|0.48|0.52|0.53|0.54|
>
> Different category:
> |Range|0.2|0.3|0.4|0.5|0.6|0.7|
> |:-|:-:|:-:|:-:|:-:|:-:|:-:|
> |Class +1|0.75|0.70|0.68|0.65|0.62|0.61|
> |Class -1|0.78|0.72|0.66|0.64|0.63|0.60|
> - These figures highlight a key pattern in the local distribution characteristics: **samples within the same class tend to show smaller value differences with nearby neighbors, whereas proximity across classes often corresponds to larger differences**.
> - We have revised the figure caption to provide more details: “*The average relative difference after min-max normalization between the value of a sample and the values of its neighbors within the same (c) and different (d) categories. The Random dataset defined in Eq. (12) is utilized...*”
>
> > **Q2: Line 152.**
>
> We have revised the sentence to “*Additional results are presented in Fig. 6 of the Appendix.*” Moreover, all appendix references have been updated to **indicate the specific section, figure, or table**.
>
> > **Q3: AME-based SV estimator.**
>
> The sentence has been revised to: “*To estimate the Shapley values of samples, we apply the AME-based Shapley value estimator [25], setting the number…*” Additionally, all relevant descriptions have been revised for improved clarity.
>
> > **Q4: Choice of $\mathcal{N} _ {k}$ and the similarity metric.**
>
> - **Section A.11 (Figs. 10(c) and (d))** evaluates the impact of neighborhood size ($k$) on IncGLOC performance. The tables below further present ablation studies of $k$ for GLOC and DecGLOC, reporting integer MSE ratios between AME and our approach. Consistent with IncGLOC, **optimal performance is observed at $k=5$ or $k=8$** across various datasets, suggesting these as recommended default settings.
>
> For GLOC:
> |$k$|3|5|8|10|15|
> |:-|:-:|:-:|:-:|:-:|:-:|
> |Electricity|27:1|**50:1**|46:1|45:1|30:1|
> |CIFAR10|53:1|**96:1**|95:1|87:1|48:1|
> |2Dplanes|58:1|**105:1**|**105:1**|101:1|64:1|
> |BBC|4:1|6:1|**8:1**|5:1|4:1|
>
> For DecGLOC:
> |$k$|3|5|8|10|15|
> |:-|:-:|:-:|:-:|:-:|:-:|
> |Electricity|39:1|**67:1**|65:1|59:1|45:1|
> |MiniBooNE|15:1|**21:1**|19:1|16:1|13:1|
> |CIFAR10|47:1|95:1|**96:1**|85:1|71:1|
> |Fried|3:1|**9:1**|**9:1**|7:1|5:1|
> - We use cosine similarity as the metric and compare it with Euclidean distance (with $\mathcal{S} _  {i,j} = d _ {E}(\boldsymbol{x} _ {i},\boldsymbol{x} _ {j}) \cdot \left[1 - 2\mathcal{I}(y _ i = y _ j)\right]$) under two settings: Shapley value estimation and mislabeled data detection. Results show that **cosine similarity consistently outperforms Euclidean distance**, likely due to its ability to capture directional alignment while ignoring magnitude, thus better reflecting semantic similarity between data points.
>
> |Metric|Electricity|MiniBooNE|CIFAR10|BBC|Fried|2Dplanes|Pol|Covertype|Nomao|Law|Creditcard|Jannis|
> |:-|:-:|:-:|:-:|:-:|:-:|:-:|:-:|:-:|:-:|:-:|:-:|:-:|
> |Cosine|**50:1**|**8:1**|**96:1**|**6:1**|**82:1**|**105:1**|**7:1**|**113:1**|**44:1**|**18:1**|**54:1**|**206:1**|
> |Euclidean|27:1|5:1|85:1|6:1|36:1|69:1|6:1|102:1|38:1|15:1|51:1|127:1|
>
> |Dataset|Pol|Jannis|Law|Covertype|Nomao|Creditcard|
> |:-|:-:|:-:|:-:|:-:|:-:|:-:|
> |Cosine|**0.66±0.009**|**0.30±0.007**|**0.96±0.008**|**0.53±0.011**|**0.68±0.006**|**0.46±0.005**|
> |Euclidean|0.61±0.006|0.28±0.004|0.95±0.009|0.52±0.010|**0.68±0.011**|0.45±0.006|
>
> > **Q5: Comparison with CS-Shapley.**
>
> Following your valuable suggestion, we extended our mislabeled data detection and low-value data addition experiments to include comparisons with CS-Shapley. As shown below, **GLOC, which accounts for both global and local value distributions, consistently outperforms CS-Shapley**. These results will be included in our camera-ready version for more comprehensive evaluation.
>
> |Dataset|Pol|Jannis|Law|Covertype|Nomao|Creditcard|
> |:-|:-:|:-:|:-:|:-:|:-:|:-:|
> |CS-Shapley|0.56±0.008|0.25±0.006|0.92±0.007|0.51±0.009|0.65±0.015|0.41±0.007|
> |GLOC|**0.66±0.009**|**0.30±0.007**|**0.96±0.008**|**0.53±0.011**|**0.68±0.006**|**0.46±0.005**|
>
> Accuracy on MiniBooNE.
> |Ratio|0.05|0.10|0.15|0.20|
> |:-|:-:|:-:|:-:|:-:|
> |GLOC|**0.660**|**0.614**|**0.529**|**0.395**|
> |CS-Shapley|0.681|0.631|0.587|0.517|
>
> Accuracy on 2Dplanes.
> |Ratio|0.05|0.10|0.15|0.20|
> |:-|:-:|:-:|:-:|:-:|
> |GLOC|**0.734**|**0.678**|**0.636**|**0.520**|
> |CS-Shapley|0.759|0.726|0.672|0.610|
>
> > **Q6: Value over- or under-estimation for new data due to reliance on existing estimates.**
>
> As you mentioned, the accuracy of original values can influence the precision of newly computed values. This limitation has been acknowledged in **Section A.12**. To mitigate this, it is recommended to apply the dynamic valuation methods on reliable initial values. Luckily, **the proposed GLOC algorithm can calibrate original values to enhance their accuracy (as shown in Table 1) and can be applied prior to IncGLOC and DecGLOC**.
>
> > **Q7: Runtime analysis.**
>
> **Fig. 9** compares the computation times of IncGLOC and DecGLOC with existing baselines. The results show that **methods avoiding Shapley value re-estimation—such as KNN, KNN+, and ours—consistently achieve superior efficiency**. Moreover, the table below compares the computation time of the proposed GLOC method (based on AME) with the original AME. Results indicate that GLOC, which introduces only two distribution-aware regularization terms to AME, **incurs minimal additional training time while improving Shapley value estimation accuracy by up to 206 times** (Table 1).
>
> |Dataset|BBC|CIFAR10|Law|Electricity|Fried|
> |:-|:-:|:-:|:-:|:-:|:-:|
> |AME|21.45s|489.54s|7.24s|8.32s|10.48s|
> |GLOC|21.58s|490.13s|7.25s|8.34s|10.53s|
>
> > **Q8: Inappropriate descriptions of “benchmark Shapley values”.**
>
> We apologize for the inappropriate wording and have corrected it to "*ground truth Shapley values*." The estimation procedure follows that outlined in **Section 3 (Lines 142-145)**, where AME estimates—derived by setting the number of sampled subsets equal to the total sample size—are considered ground truth. This configuration theoretically ensures that the estimated values closely approximate the true Shapley values due to the large number of subsets [1].
>
> > **Q9: Further limitations.**
>
> Thanks for your thoughtful feedback. Below, we provide a detailed analysis of each potential limitation you outlined.
>
> 1. Compared to AME, which employs L1 regularization, our approach adopts L2 regularization, thereby reducing reliance on the sparsity assumption inherent in AME.
> 2. Although our method introduces the hyperparameter $𝑘$ to control the neighborhood range, experimental results show that default values of 5 or 8 consistently yield satisfactory performance (see Q4 for details).
> 3. Very insightful! We fully agree that neighborhood dependence may pose privacy risks, and have incorporated this concern in the limitations section. Nevertheless, these risks can be partially mitigated through differential privacy or local perturbation techniques, presenting a promising avenue for future research.
> 4. As discussed in response to Q6 and our limitations section, the effectiveness of dynamic valuation methods can indeed be influenced by the quality of initial values. Fortunately, the GLOC algorithm can first enhance low-quality initial values before applying IncGLOC and DecGLOC for dynamic valuation.
> 5. This limitation is common to almost all valuation methods based on utilities. Our approach imposes no restrictions on model or utility selection, as well as ensuring compatibility with various existing valuation techniques.
> 6. For reasons of efficiency and adaptability, many practical valuation methods do not strictly satisfy the Shapley efficiency axiom. Nonetheless, extensive experiments demonstrate the strong practical utility of our approach. Our future research will focus on developing methods that better meet these criteria while maintaining computational efficiency.
>
> Accordingly, the following content has been added to our limitation section: “*The reliance of our approach on neighbors may compromise privacy guarantees. Future work could explore privacy-preserving extensions... Moreover, our method introduces several additional hyperparameters, such as neighborhood range and regularization coefficients. While we provide recommended values and stable ranges, specific scenarios... Furthermore, despite the strong practical utility of our approach, like AME and other scalable valuation frameworks, our method does not strictly enforce the Shapley efficiency axiom. Developing approaches that better align with this axiom while...*”
>
> [1] Measuring the effect of training data on deep learning predictions via randomized experiments.

---

> > ### Comment · Reviewer_58ya · 2025-08-03
> >
> > The authors have addressed all the questions I had. Their explanations regarding the limitations are detailed, and the new results, particularly those related to computational expenses and comparative analyses are sufficient and address the questions I had. I have also reviewed the comments from other reviewers and found the authors’ rebuttals to be equally detailed and specific to questions asked. I have therefore decided to raise my score, with the expectation that the authors incorporate the new evaluations and all the rebuttal changes into the final version of the paper.

---

> > > ### Author Response · Authors · 2025-08-03
> > > **Thank you for your acknowledgment.**
> > >
> > > Dear Reviewer 58ya,
> > >
> > > Thank you very much for your timely reply! We are sincerely grateful that you found our revisions and explanations to be sufficiently comprehensive in addressing your concerns. We also greatly appreciate your recognition of our efforts in providing detailed and specific responses to the comments raised by all reviewers.
> > >
> > > Your decision to raise the score for our submission is greatly appreciated, truly encouraging, and means a great deal to us. We will carefully incorporate all the changes and evaluations made during the rebuttal phase into the final version of the manuscript, as they significantly contribute to improving the quality and clarity of our work. Once again, we sincerely thank you for your thoughtful feedback and for the significant role you have played in helping us improve the quality of our work!
> > >
> > > Kind regards,
> > >
> > > Authors

---

### Official Review · Reviewer_4jgi · 2025-07-02

**Clarity:** 4
**Significance:** 3
**Originality:** 3
**Rating:** 4
**Confidence:** 4

**Summary:**

This paper introduces a novel approach to data valuation based on the previously proposed AME method. There are two major improvements: (1) as observed in experiments, the distribution of data Shapley is more close to Gaussian distributions, therefore, they replace the Laplace priors with Gaussian priors; (2) they also observe that data points with the same label in the neighborhood usually have similar values, so they add a regularization term to enforce neighborhood consistency to refine Shapley value estimation. Additionally, the authors present dynamic data valuation methods that update data values without recomputing Shapley values when new data are added. They evaluate their approach on tasks like Shapley value estimation, mislabeled data detection, and dynamic data valuation.

**Questions:**

Could the authors address Weakness 2 and Weakness 3 in their response?

**Ethical Concerns:**

["NO or VERY MINOR ethics concerns only"]

**Final Justification:**

The authors' responses addressed my main concerns. The experimental results demonstrate the effectiveness of their approach.  However, the work still appears somewhat incremental to me. The distributional aware approach, (i.e. the change of the regularization) does not really constitute a fundamentally novel approach to me.

**Limitations:**

Yes.

**Paper Formatting Concerns:**

None.

**Quality:**

3

**Strengths And Weaknesses:**

**Strengths:**
- The paper presents insightful empirical findings on the distribution of Shapley values across data points and introduces a novel method that leverages these observations to enhance existing data valuation approaches.
- The proposed method is simple and scalable, and it appears to be effective across different tasks.
- The experiments are comprehensive, covering diverse datasets and comparing against multiple baselines. The paper provides clear implementation details.

**Weaknesses:**
1. While the method appears to be effective, it is an extension of AME rather than a fundamentally new approach. The core novelty lies in the regularization terms (Gaussian prior + neighborhood consistency), which, while well-justified, do not represent a paradigm shift in data valuation.
2. The MSE evaluation uses AME-estimated Shapley values as the "ground truth", which may favor AME-based methods (including the proposed methods) over non-AME baselines (e.g., KNN Shapley, Data Shapley). In addition,  the MSE of all methods appear to be very small, it’s unclear whether the improvements are practically significant or if alternative ground truth estimations (e.g., Monte Carlo Shapley) would yield different conclusions. A sensitivity analysis using different ground truth methods would strengthen the validity of the comparisons.
3. In the data addition/removal experiments, it is not clear whether the rapid accuracy changes stem from: (1) improved Shapley value estimation accuracy, or (2) the neighborhood consistency regularization forcing similar data points to have similar values, causing them to be removed together and resulting in a faster performance drop.

---

> ### Author Rebuttal · Authors · 2025-07-31
>
> Dear Reviewer 4jgi,
>
> We sincerely appreciate your thoughtful review and valuable feedback. Thank you for recognizing the novelty and scalability of our method, the depth of our empirical insights, the thoroughness of our experimental evaluation, the clarity of our implementation details, and the strength of our results. Below, we provide our detailed responses to each of your concerns.
>
> > **Q1: While the method appears to be effective, it is an extension of AME rather than a fundamentally new approach. The core novelty lies in the regularization terms, which, while well-justified, do not represent a paradigm shift in data valuation.**
>
> We sincerely appreciate your recognition of the effectiveness and soundness of our approach and humbly respond to your concerns from the following perspectives:
> - **Our approach introduces a distribution-aware perspective for data valuation, a technical path largely overlooked in prior work**. While we primarily demonstrate the effectiveness of our approach by combining it with AME under the conventional data valuation setting, **our framework is general and can be seamlessly integrated with other valuation methods**. Specifically, the proposed distribution-aware regularization terms can be applied to optimize data values either jointly with the original valuation method or as a post-processing step. As shown in **Appendix A.7** (Table 4, also presented below), combining our method with existing approaches (denoted with †) consistently yields substantial performance gains, underscoring the importance of incorporating value distribution information into valuation strategies.
> |Dataset|Pol|Jannis|Law|Covertype|Nomao|Creditcard|
> |:-|:-:|:-:|:-:|:-:|:-:|:-:|
> |KNN Shapley|0.28±0.003|0.25±0.004|0.45±0.014|0.51±0.021|0.47±0.013|0.43±0.004|
> |KNN Shapley†|**0.73±0.007**|**0.33±0.006**|**0.96±0.011**|**0.55±0.016**|**0.70±0.012**|**0.50±0.006**|
> |Data Shapley|0.50±0.011|0.23±0.003|0.94±0.003|0.37±0.004|0.65±0.005|0.36±0.006|
> |Data Shapley†|**0.77±0.010**|**0.31±0.005**|**0.97±0.008**|**0.51±0.006**|**0.72±0.008**|**0.48±0.008**|
> |Beta Shapley|0.46±0.010|0.24±0.003|0.94±0.003|0.41±0.003|0.66±0.005|0.43±0.005|
> |Beta Shapley†|**0.75±0.009**|**0.30±0.008**|**0.97±0.007**|**0.54±0.005**|**0.74±0.007**|**0.49±0.007**|
> - Dynamic data valuation remains a largely underexplored yet practically significant challenge. Existing data valuation methods, such as AME, require re-estimating Shapley values over the entire dataset whenever data changes occur, which is computationally expensive, particularly in the context of rapidly expanding data volumes. To address this, **we propose two dynamic data valuation approaches that are agnostic to specific valuation methods**. These approaches enable efficient optimization for new values based solely on the current dataset and previously computed values, eliminating the need for recalculating the Shapley values.
>
> In summary, our approach is not limited to AME and can be seamlessly integrated with a wide range of existing data valuation methods to further refine data values.
>
> > **Q2: Fairness for using AME-estimated Shapley values as ground truth in MSE evaluation and the need for experiments with alternative ground truth estimations.**
>
> Good question! We humbly respond to your concerns from the following aspects:
> - To approximate the ground-truth Shapley values, we employ the AME method with the number of sampled subsets set equal to the total number of data points. **This configuration ensures a high-fidelity approximation, as a sufficiently large number of subsets theoretically guarantees convergence to the true Shapley values [1]**.
> - We adopt MSE as the evaluation metric in two experiments. A lower MSE indicates that the estimated values more closely approximate the true Shapley values, reflecting improved data valuation accuracy. The first experiment assesses the accuracy of data valuations generated by AME and our proposed GLOC method in approximating the true Shapley values. Given that both methods build upon the AME framework, **the comparison is inherently fair**. As shown in Table 1, **GLOC consistently yields valuations that more closely align with the ground-truth Shapley values than those produced by AME**.
> - In the second experiment, we employ MSE to assess the performance of various dynamic data valuation methods. Among the baselines, Monte Carlo Shapley (MC) and Truncated Monte Carlo Shapley (TMC) estimate the Shapley value of each sample in the new dataset from scratch, whereas the other three methods—Delta, KNN, and KNN+—infer new values based on previously computed estimates. As previously noted, our two dynamic data valuation approaches are method-agnostic. Therefore, **to ensure fair comparison, all methods (including ours), except MC and TMC, are initialized with prior MC estimates**. To better clarify this, we have added the following sentence to **Line 340** of the manuscript: “*Except for MC and TMC, which estimate Shapley values from scratch, all other compared methods are initialized with MC-estimated data values.*” Furthermore, inspired by your valuable suggestion, **we conduct additional experiments where the ground-truth Shapley values were also estimated using MC**. From the results below, our method continues to achieve superior performance even under this revised evaluation protocol.
> |Manner|Add|Add|Add|Add|Remove|Remove|Remove|Remove|
> |:-|:-:|:-:|:-:|:-:|:-:|:-:|:-:|:-:|
> |Dataset|Electricity|MiniBooNE|CIFAR10|Fried|Electricity|MiniBooNE|CIFAR10|Fried|
> |MC|8.78e-6|9.43e-6|2.67e-5|9.62e-6|5.45e-6|3.44e-6|8.72e-6|6.47e-6|
> |TMC|7.87e-5|5.67e-5|1.35e-4|9.57e-6|2.28e-5|3.21e-5|9.46e-5|4.58e-5|
> |Delta|5.45e-6|2.53e-6|6.67e-6|2.37e-6|9.74e-6|1.44e-5|8.32e-6|9.78e-6|
> |KNN|9.07e-6|3.49e-6|9.37e-6|9.56e-6|4.88e-6|4.27e-6|3.39e-6|1.37e-5|
> |KNN+|9.11e-6|1.28e-5|1.01e-5|3.44e-6|1.21e-6|4.58e-6|1.05e-5|2.49e-5|
> |Ours|**1.21e-6**|**1.45e-6**|**2.03e-6**|**7.63e-7**|**6.56e-7**|**1.04e-6**|**8.88e-7**|**2.02e-6**|
> - Although the absolute MSE values are small, our method demonstrates substantial relative improvements over baseline approaches. To illustrate this, **Table 1** in the manuscript presents the MSE ratio between AME and our method, **with the most pronounced improvement reaching up to 206-fold**. Additionally, the table below reports the MSE ratios (rounded to integers) between baseline methods and our approach under the dynamic data valuation setting, where AME estimates serve as a proxy for the true Shapley values and MC estimates are used as initial values. From the results, **our method achieves at least a 2× and up to a 506× improvement in Shapley value estimation accuracy compared to other baseline approaches**.
> |Manner|Add|Add|Add|Add|Remove|Remove|Remove|Remove|
> |:-|:-:|:-:|:-:|:-:|:-:|:-:|:-:|:-:|
> |Dataset|Electricity|MiniBooNE|CIFAR10|Fried|Electricity|MiniBooNE|CIFAR10|Fried|
> |MC|33:1|40:1|14:1|12:1|8:1|3:1|19:1|5:1|
> |TMC|506:1|63:1|149:1|6:1|47:1|30:1|226:1|151:1|
> |Delta|4:1|2:1|3:1|2:1|41:1|18:1|11:1|6:1|
> |KNN|22:1|3:1|7:1|25:1|8:1|3:1|3:1|19:1|
> |KNN+|20:1|23:1|16:1|3:1|3:1|3:1|15:1|20:1|
>
> All these experiments will be incorporated into the camera-ready version of our manuscript. Once again, we sincerely appreciate your valuable feedback, which has helped improve the comprehensiveness of our evaluation.
>
> > **Q3:  Causes of rapid accuracy changes in data addition/removal experiments: improved Shapley value estimation vs. neighborhood consistency regularization.**
>
> We humbly respond to your concerns from the following two points:
> - **The incorporation of the proposed neighborhood consistency regularization can enhance the accuracy of Shapley value estimation**. This is evidenced by the ablation studies in Fig. 3, where the removal of the local regularization term results in a marked degradation in estimation performance (i.e., increased MSEs). To further validate this, we conduct additional ablation experiments on a broader range of datasets. As shown below, the results consistently confirm the effectiveness of the regularization term in improving the accuracy of Shapley value estimation. This enhanced accuracy, in turn, enables our method to more effectively distinguish between high- and low-quality samples, leading to greater impact on model performance during value-based data addition and removal.
> |Dataset|CIFAR10|Fried|2Dplanes|Covertype|Creditcard|Jannis|
> |:-|:-:|:-:|:-:|:-:|:-:|:-:|
> |GLOC|**96:1**|**82:1**|**105:1**|**113:1**|**54:1**|**206:1**|
> |$-\mathcal{R}_{g}$|6:1|21:1|34:1|14:1|20:1|68:1|
> |$-\mathcal{R}_{l}$|5:1|17:1|12:1|9:1|22:1|35:1|
> - Moreover, we conduct ablation studies under the value-based point addition and removal setting, where the global  $\mathcal{R} _ {g}$ and local $\mathcal{R} _ {l}$ regularization terms are independently removed. From the results, **removing either term leads to a slower rate of performance degradation**. If rapid accuracy changes are primarily driven by neighborhood consistency regularization, which causes similar samples to be removed together, removing the global term should accelerate the performance drop. Thus, we conclude that **the primary driver of the rapid accuracy change is the enhanced Shapley value estimation, facilitated by the synergistic effect of both global and local regularization terms**.
>
> Accuracy on the Fried dataset with varying ratios of data removed.
> |Ratio|0.2|0.4|0.6|0.8|
> |:-|:-:|:-:|:-:|:-:|
> |GLOC|**0.786**|**0.764**|**0.763**|**0.742**|
> |$-\mathcal{R}_{g}$|0.790|0.781|0.769|0.743|
> |$-\mathcal{R}_{l}$|0.792|0.776|0.774|0.751|
> |AME|0.809|0.799|0.785|0.768|
>
> Accuracy on the MiniBooNE dataset with varying ratios of data added.
> |Ratio|0.05|0.1|0.15|0.2|
> |:-|:-:|:-:|:-:|:-:|
> |GLOC|**0.658**|**0.613**|**0.527**|**0.394**|
> |$-\mathcal{R}_{g}$|0.662|0.615|0.549|0.428|
> |$-\mathcal{R}_{l}$|0.680|0.622|0.560|0.451|
> |AME|0.714|0.647|0.640|0.637|
>
> [1] Measuring the effect of training data on deep learning predictions via randomized experiments. ICML 2022.

---

> > ### Comment · Reviewer_4jgi · 2025-08-03
> >
> > Thank you for the detailed responses. I have no further questions. The response slightly raised my evaluation. However, the work still appears somewhat incremental to me. The distributional aware approach, (i.e. the change of the regularization) does not really constitute a fundamentally novel approach to me.

---

> > > ### Author Response · Authors · 2025-08-03
> > > **Thank you for your prompt reply.**
> > >
> > > Dear Reviewer 4jgi,
> > >
> > > Thank you very much for your prompt reply! We are delighted that you found our responses have adequately addressed your concerns and raised your evaluation. We are truly grateful for all your valuable comments and constructive suggestions, which have significantly enhanced the quality of our manuscript—particularly in terms of improving the completeness and rigor of our experimental evaluation.
> > >
> > > As existing data valuation methods have largely overlooked the exploration of data value distribution, our proposed global and local characteristics-based regularization approach represents a relatively general framework that can be effectively integrated with many existing valuation techniques, especially since regularization is a highly valuable optimization strategy in machine learning. Moreover, data value calibration from the perspective of value distribution remains an underexplored area, and we believe our work holds the potential to inspire a range of new methodologies in this direction. In fact, our current manuscript introduces two novel approaches for dynamic data valuation derived from this very perspective.
> > >
> > > Once again, we would like to express our heartfelt appreciation for your thoughtful review and the valuable role you have played in improving our work!
> > >
> > > Warm regards,
> > >
> > > Authors

---

### Note · Authors · 2025-08-12

Dear area chairs and reviewers,

We sincerely appreciate your significant time and effort invested throughout the review process, as well as your insightful and valuable feedback, which has been instrumental in refining and enhancing our manuscript.

We are particularly grateful that **all reviewers found our responses to be detailed and constructive, effectively addressing their concerns**, and we deeply value their positive recognition of our contributions. The reviewers found **our method to be novel, valuable, and scalable** (Reviewers 4jgi, 58ya, Vwbk, and 92n8), **our findings regarding value distribution to be insightful and interesting** (Reviewers 4jgi, 58ya, and Vwbk), **the experiments to be comprehensive and diverse** (Reviewers 4jgi, 58ya, and 92n8), **the performance to be strong and promising** (Reviewers 4jgi and 92n8), and **the paper to be well-written and easy to understand** (Reviewer 58ya). We will carefully incorporate all the clarifications and experiments made during the rebuttal phase into the final version of the manuscript, as they significantly contribute to improving the quality and clarity of our work.

We sincerely hope that the constructive feedback and overall positive consensus among all reviewers, together with the clarifications and improvements we have made during the rebuttal phase, will contribute to a favorable decision. Once again, we wish to convey our sincere gratitude to all of you for your invaluable contributions to our work.

Kind regards,

Authors

---

### Decision · Program_Chairs · 2025-09-17

**Decision:**

Reject

**Comment:**

The paper proposes a variant of AME for Shapley value estimation in data valuation problems. It shows empirically that the prior on data values is closer to Gaussian than Laplace and proposes to change the regularizer accordingly, also adding local information. The paper empirically shows the efficiency of this proposed method, on multiple datasets and tasks. It also considers the dynamic scenario with addition/removal of data (which was not much studied yet).

All reviewers had a globally positive opinion of the overall results. Notably, after the rebuttal, all reviewer were convinced by the strength of the empirical evaluation showing that the proposed method is efficient.

However, all reviewers also agreed that the method proposed is incremental (a change of regularizer in AME) and that the technical novelty is limited. Unfortunately, given the high pressure at NeurIPS this year, we found that the paper indeed does not meet the bar in terms of novelty and must therefore recommend its rejection.